# Accelerated growth increases the somatic epimutation rate in trees

Ming Zhou [1,7], Gerhard Schmied [2,7], Binh Thanh Vo[1], Monika Bradatsch[2], Giulia Resente [3,4], Rashmi Hazarika[1], Ioanna Kakoulidou [1], Maria-Cecília Costa [1], Michele Serra [1], Richard L. Peters[2], Enno Uhl[5], Robert J. Schmitz [6], Torben Hilmers [2], Astor Toraño Caicoya [2], Alan Crivellaro [3], Hans Pretzsch [2] ✉ & Frank Johannes[1] ✉

Trees are integral to ecosystems and hold considerable economic importance. Their exceptional longevity and modular structure also make them valuable models for studying the long-term accumulation of somatic mutations and epimutations in plants. Empirical evidence indicates that the annual rate of these stochastic events correlates negatively with generation time, suggesting that species with long lifespans have evolved mechanisms to mitigate the build-up of deleterious somatic variants. It has been hypothesized that this reduction is achieved by slowing growth and minimizing the number of cell divisions per unit time, thereby reducing errors associated with DNA replication. However, a direct test of this "mitotic-rate hypothesis" remains technically challenging. Here we take advantage of a 150 year-old experiment in European beech to show that a thinning-induced growth acceleration increases the annual rate of somatic epimutations in main stems and lateral branches of trees. We demonstrate that this effect is accompanied by a proportional increase in the rate of cell divisions per unit time. These findings support the notion that life-history constraints on growth rates in trees are not merely a trade-off between resource allocation and structural stability but also a strategy to preserve genetic and epigenetic fidelity over extended lifespans.

Trees are among the longest-living organisms on earth. They have critical ecosystem functions, and continue to be of major economic importance[1]. Perhaps due to their longevity, sessile life-style and modular nature, many tree species have evolved a remarkable degree of phenotypic plasticity in response to environmental stressors[2]. These plastic responses are at least in part driven by epigenetic mechanisms, including DNA methylation and histone modifications[3]. One example is priming in Norway spruce (*Picea abies*), where hormonal exposure of seedlings can induce an epigenetically-encoded stress memory that confers more effective resistance to insect attacks later in life[4]. Although such epigenetic memories are typically lost with passage into the next generation[3], their ecological consequences are nonetheless relevant, given the ontogenetic time-scales involved[5].

In addition to transient epigenetic effects, there are also more stable epigenetic changes that occur stochastically during development and aging[6,7]. A well-characterized form of such changes is accidental gains and losses of DNA cytosine methylation, a phenomenon that has been termed "spontaneous epimutation"[6]. Epimutations at CG

[1]Plant Epigenomics, Technical University of Munich, Freising, Germany. [2]Tree Growth & Wood Physiology, Technical University of Munich, Freising, Germany. [3]Department of Agricultural, Forest and Food Sciences, University of Torino, Grugliasco (TO), Italy. [4]Department of Botany and Landscape Ecology, University of Greifswald, Greifswald, Germany. [5]Bavarian State Institute of Forestry (LWF), Bavarian State Ministry of Food, Agriculture and Forestry, Freising, Germany. [6]Department of Genetics, University of Georgia, Athens, GA, USA. [7]These authors contributed equally: Ming Zhou, Gerhard Schmied. ✉e-mail: hans.pretzsch@tum.de; f.johannes@tum.de

dinucleotides in plants are particularly important because they are inherited not only during somatic cell division (mitosis), which allows the changes to be passed on within an individual, but also during the formation of reproductive cells (meiosis), ensuring that the changes can be passed on to the next generation[8–20]. It is believed that they originate mainly from errors made by CG methyltransferases during the maintenance of DNA methylation at cell division[19]. When such epimutations occur in the shoot apical meristem (SAM) - a small population of stem cells at the shoot apex - they often become fixed in the cell lineages that differentiate into aerial structures such as stems and branches[21]. This fixation results from somatic drift driven by pre-cursor cell sampling during organ formation[22]. Somatically fixed epi-mutations therefore appear at high frequencies in many plant tissues, and can be detected using bulk sequencing approaches[17] (Methods).

Recent evidence shows that the yearly rate of fixed somatic epi-mutations correlates negatively with generation time, being about ~2 orders of magnitude lower in long-lived trees (e.g. *Populus tricocarpa*: ~$0.08 \times 10^{-4}$, CG per year) compared to annual plant species (e.g., *Arabidopsis thaliana*: ~$9.3 \times 10^{-4}$ per CG per year)[23]. Similar observations have been made at the level of somatic genetic mutations[13,24–31]. This rate reduction may serve as a protective mechanism in long-lived species to delay the accumulation of potentially deleterious (epi) mutations during aging[32]. Multiple lines of evidence indicate that this is achieved by slowing growth and the number of stem cell divisions per unit time[22,33,34]. However, experimental tests of this hypothesis remain challenging, as they require a direct manipulation of these develop-mental parameters coupled with a long-term assessment of their epi-mutational consequences. Moreover, in vivo measurements of stem cell division would need to rely on live imaging of intact meristems in mature trees - an approach that remains technically challenging.

Here, we overcome many of these challenges and provide a test of the "mitotic-rate hypothesis" applied to mature trees growing under natural conditions. Using European beech (*Fagus sylvatica* L.) as a model, we take advantage of one of the oldest continuously measured experimental plots in the world. The plot contains an even-aged beech stand that was established in 1822 and monitored for growth at regular intervals until the present. Starting about 150 years ago, different thinning strategies were applied to subplots of this experiment, which has resulted in differences in the average stem growth rates of trees among subplots. We show that thinning-induced growth acceleration significantly increases the annual rate of somatic epimutations in both stems and lateral branches, and that this effect is accompanied by a proportional increase in the rate of cell divisions of SAM-derived cell lineages. Our results lend support to the "mitotic-rate hypothesis" as a key explanation for the delay in (epi)mutational meltdown in long-lived plants. Since somatic CG epimutations can be inherited across generations, our work further illustrates how developmental processes within individual trees can impact epigenetic diversity at the popula-tion level over extended timescales.

## Results

### Experimental stand thinning leads to accelerated growth

The Fabrikschleichach 15 (FAB 15) long-term thinning experiment in European beech is one of the oldest continuously monitored experi-mental plots (Methods). The ~0.4 ha sized unthinned, moderately thinned, and heavily thinned plots of FAB 15 were established in 1870/1871 in a 48-year-old, even-aged European beech stand that originated from natural regeneration by shelterwood cutting of a beech forest in 1822[35]. The managed plots were thinned 12 times at regular intervals since their establishment, with a significant impact on stand growth dynamics[36]. In 2020, 138 trees remained in the unthinned, 70 in the moderately thinned, and 40 in the heavily thinned plot. This resulted in different average stem growth rates of 8.1 cm² yr⁻¹ (unthinned), 12.2 cm² yr⁻¹ (moderate), and 16.4 cm² yr⁻¹ (heavy) within the period from 1822 to 2020, which differed significantly from each other

($P \leq 0.0001$ for all three comparisons with pairwise Wilcoxon rank sum test and Bonferroni correction) (Supplementary Fig. 1).

### Accelerated growth is accompanied by increased cell counts in stems

We felled two representative trees from the moderately thinned plot (tree 109) and the heavily thinned plot (tree 171) to further study the impact of differential thinning on growth characteristics (Fig. 1a, b). The stem growth rates of these two trees were 12.2 cm² yr⁻¹ (tree 109, moderately thinned) and 21.8 cm² yr⁻¹ (tree 171, heavily thinned). By the time of the last measurement in 2020, tree 171 was only slightly taller than tree 109 (42.8 m vs. 42.4 m), but was ~1.3 times thicker at diameter breast height (DBH at 1.3 m: 72.3 cm vs. 55.1 cm) (Fig. 1c, Supplementary Fig. 1), and had a 2.83 fold larger crown (crown pro-jection area: 102 m² vs. 36 m²). This indicates that tree 171 had sig-nificantly more foliage and branches, despite both trees germinating in the same year. The differential stem growth rates were reflected in the historical growth trajectories of both trees (Fig. 1d). To evaluate whe-ther differences in stem growth rate are the result of cell size expan-sion or increased cell proliferation, we performed cell count assays of xylem vessels, fiber cells, and parenchyma cells in stem discs (Meth-ods). The cumulative number of cells was highly correlated with the cumulative growth curve of the trees ($r = 0.96$). On average, tree 171 had significantly more and larger cells per annual ring than tree 109 ($P \leq 0.0001$, Fig. 1e–g, Supplementary Fig. 2), suggesting that growth acceleration is accompanied by an increased cell division rate per unit time.

### Accelerated growth affects epimutation accumulation in the main stem

Cells in the outer cambium are the endpoints of cell lineages that have their origin in a few SAM-derived ancestor cells at the stem core[37–39]. The above cell count data imply that the depth of these cell lineages is higher in tree 171 than in tree 109. Under the assumption that spon-taneous epimutations arise with every cell division, we expected methylation divergence between cambium cells sampled from oppo-site sides of the stem to be higher in trees from the heavily thinned compared to the moderately thinned plot (Fig. 2a). To test this hypothesis, we obtained cambium samples from additional trees, four trees from the moderately thinned and three from the heavily thinned plot (Methods). For each tree, three cambium samples were collected: two neighboring replicate samples from the same side of the stem and one sample from the polar opposite side (Fig. 2a). The replicates were treated as controls, as their methylation divergence is expected to be minimal due to shared cell lineage ancestry close to the cambium[40,41] (Fig. 2a). We generated whole genome bisulfite sequencing (WGBS) data for all samples ($N = 18$ WGBS samples in total)(Methods). Con-sistent with our hypothesis, methylation divergence between samples from opposite sides of the stems was 2.64-fold higher in trees from the heavily thinned plot compared to trees from the moderately thinned plot (Fig. 2b, c). By contrast, methylation divergence between replicate samples was much lower and not significantly different between plots. These findings indicate that accelerated growth, accompanied by increased cell division rates, is associated with a significant increase in epimutation accumulation in the main stem.

### Accelerated growth affects epimutation accumulation along lateral branches

Assuming that the relationship between growth rate, cell divisions, and epimutations is a broad effect across the tree, not limited to the main stem, we expected to observe a similar pattern in the lateral branches throughout the crown. The cell lineages that initiate leaf formation on distal terminal branches derive from a common ancestor stem cell approximately at the last shared branch point[22]. Leaves that are more distal from each other in the branching topology are separated by

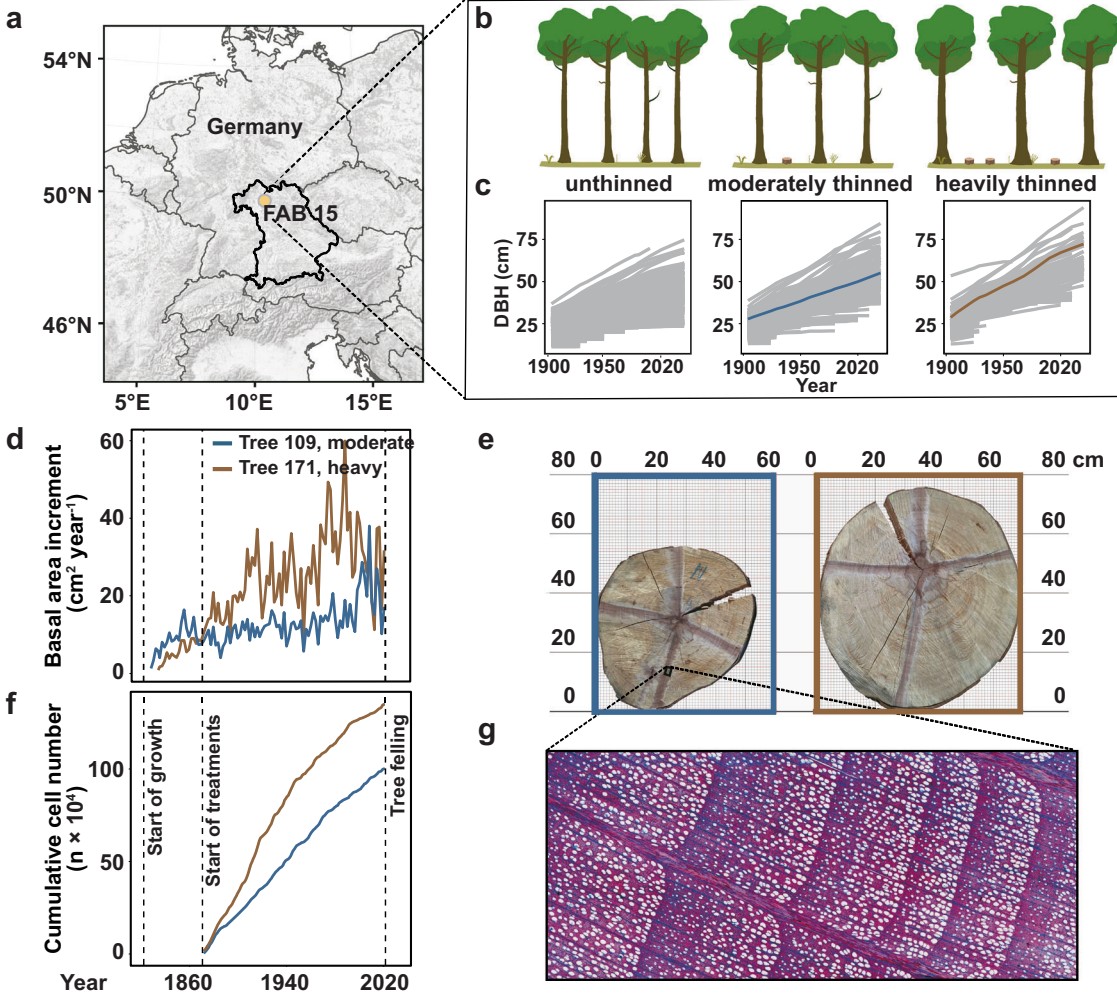

**Fig. 1 | Map of sampling site and the tree growth. a** Overview of the location and **b** treatment design of the FAB15 long-term thinning experiment in European beech. **c**, **d** We relied two representative sample trees from the moderately (Tree 109) and heavily thinned (Tree 171) plots. The different thinning intensities resulted in different tree growth trajectories and growth rates. **e**–**g** Wood anatomical analyses showed Tree 171 had a higher cell division rate than Tree 109. Source data are provided as a Source Data file.

more stem cell divisions[22] and therefore expected to have accumulated more epimutations between them[17]. One way to assess this is to employ an intra-organismal phylogenetic method that relates DNA methylation divergence among distal leaves to their pairwise branching distance (in years)[17] (Fig. 3a). Application of this method to tree 171 and tree 109 should reveal growth-related differences in epimutation accumulation along lateral branches per unit time. To test this, we first generated whole-genome bisulfite sequencing (WGBS) data from leaf samples collected from tree 171 ($N = 10$ WGBS samples) and tree 109 ($N = 7$ WGBS samples). The leaves were carefully selected to provide a broad representation of the three-dimensional branching architecture of each tree (Fig. 3a).

To be able to detect epimutations that emerge at individual CG dinucleotides as well as in larger (~100–200 bp) genomic regions, we identified single methylation polymorphisms (SMPs) and differentially methylated regions (DMRs) between leaf samples using jDMR[12] (Methods). Unsupervised clustering of the samples based on their SMP or DMR profiles recapitulated the known branching topology of each tree (Fig. 3a, Supplementary Fig. 3 and Supplementary Data 2), indicating that our SMP/DMR calling approach was successful at reconstructing the branching (i.e., cell lineage) history of the detected CG epimutations. In addition to DNA methylation profiling, we also determined the ages of the sampled branches from branch disks

(Methods). Together, these data allowed us to calculate DNA methylation divergence as a function of divergence time between all leaf pairs (Fig. 3b, c). Consistent with recent work in *P. trichocarpa*[13,17], CG methylation divergence increased gradually with divergence time in both trees. This was true at the level of individual CGs as well as at the level of regions (Fig. 4d). However, divergence was visibly more rapid in tree 171 than in 109 (Fig. 4d), suggesting that tree 171 accumulated epimutations at a faster rate per unit time.

To obtain direct estimates of these rates, we employed AlphaBeta[17,42]. At the genome-wide scale we found that the spontaneous mCG gain rate (α) and loss rate (β) were ~1.22 times higher in tree 171 than in 109 on average (tree 171: α = $4.69 \times 10^{-6}$ (SE = $0.38 \times 10^{-6}$), β = $6.50 \times 10^{-6}$ (SE = $0.52 \times 10^{-6}$) per CG per haploid genome per year; tree 109: α = $3.83 \times 10^{-6}$ (SE = $0.41 \times 10^{-6}$), β = $5.36 \times 10^{-6}$ (SE = $0.57 \times 10^{-6}$) per site per haploid genome per year (Fig. 4b, Supplementary Data 3). A similar picture emerged for region-level epimutation rate estimates (tree 171: α = $6.79 \times 10^{-6}$ (SE = $0.46 \times 10^{-6}$), β = $9.45 \times 10^{-6}$ (SE = $0.64 \times 10^{-6}$) per 100 bp region per haploid genome per year; tree 109: α = $4.99 \times 10^{-6}$ (SE = $0.74 \times 10^{-6}$), β = $6.96 \times 10^{-6}$ (SE = $1.03 \times 10^{-6}$) per 100 bp region per haploid genome per year (Fig. 4b, Supplementary Data 3). Interestingly, despite these epimutation rate differences, genome-wide steady-state CG methylation levels were very similar between the two trees (Fig. 4a, c). This observation can be attributed to the nearly

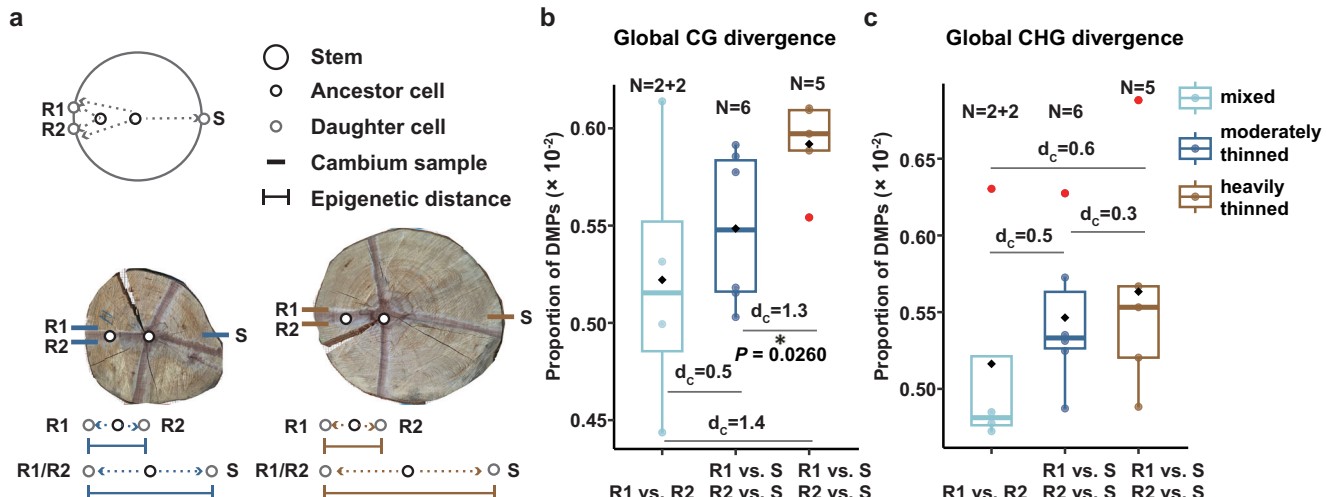

**Fig. 2 | Comparison of methylation divergence in stems with different growth rates. a** Schematic for stem sampling. Stems come from two thinning intensity treatments, one moderately thinned and the other heavily thinned. We collected four stems from moderately thinned, three stems from heavily thinned. Two neighbouring samples were collected on one side of the stem as duplicate samples (R1 and R2), and then one sample (S) was collected on the opposite side. **b, c** Comparing methylation divergence of stems from the two treatments. The x-axis represents three types of methylation divergence comparisons: R1 vs. R2 (RR) indicates the methylation divergence between two replicates on the same side of the stem. R1 vs. S and R2 vs. S represent the divergence between opposite sides of the stem. Each point in the plot corresponds to the genome-wide methylation divergence under the respective comparison, thus, N represents the number of pairwise comparisons. Stems from the heavily thinned treatment accumulate more **b** CG methylation divergence and **c** CHG methylation divergence. **b, c** Although not all comparisons reached statistical significance, the relatively large Cohen's $D$ ($d_C$) effect sizes support an increasing trend from RR to moderately thinned treatment to heavily thinned treatment. Center line, median; box limits, upper and lower quartiles; whiskers, 1.5x interquartile range. The black square is the mean, and the red dots represent potential outliers. Statistical significance was determined using an unpaired one-sided Wilcoxon rank-sum test (alternative hypothesis: true difference in means is greater than 0; $W = 26$, $P = 0.02764$). Full statistical results for all comparisons, including non-significant differences, are provided in Supplementary Data 5. *$P < 0.05$; $0.2 < d_C < 0.5$, small effect; $d_C > 0.5$, medium effect; $d_C > 0.8$, large effect. Source data are provided as a Source Data file.

proportional increase in gain and loss rates in tree 171 relative to 109, which theory predicts to result in unchanged steady-state methylation[10,17,43] (Methods).

The above genome-wide epimutation rate analysis may have included subsets of CG dinucleotides that are redundantly targeted by de novo methylation pathways[44]. The methylation gain and loss dynamics underlying our rate estimates may therefore be independent of the number of cell divisions along the branches. We sought to test if our conclusions still hold when focusing on CG sites that are exclusively targeted by METHYLTRANSFERASE 1 (MET1), whose maintenance activity is known to be restricted to DNA replication[44]. CG sites within gene body methylated (gbM) genes provide an excellent framework to test this. In plants, gbM genes are an evolutionarily conserved subset of genes that feature high CG methylation and virtually no non-CG methylation (i.e., methylation in context CHG and CHH (where H = A, T, C))[45-48]. These genes are enriched in housekeeping functions, are moderately expressed, and display low transcriptional variability across cells and tissues[49-52] (extensively reviewed in Refs. 53,54). We identified a liberal set of ~10,000 gbM genes out of the 65,000 annotated genes and pseudogenes in the current European Beech (*Fagus sylvatica* L.) reference assembly (Methods). Using CGs extracted from gbM genes, we repeated our epimutation rate estimation. We found that the rate difference between trees 171 and 109 became even more pronounced, with CG epimutation rates being ~1.66 times higher in tree 171 than in tree 109 on average (tree 171: α = 9.66 × $10^{-6}$ (SE = 0.96 × $10^{-6}$), β = 5.62 × $10^{-6}$ (SE = 0.56 × $10^{-6}$) per CG per haploid genome per year; tree 109: α =5.94 × $10^{-6}$ (SE = 1.02 × $10^{-6}$), β = 3.32 × $10^{-6}$ (SE = 0.57 × $10^{-6}$) per site per haploid genome per year) (Fig. 4b, Supplementary Data 3). These results provide further, albeit indirect, evidence that the epimutation rate differences are likely coupled with elevated cell division rates in the SAM and/or in cell lineages leading up to the formation of leaf primordia.

## Discussion

The "mitotic-rate hypothesis" proposes that the accumulation of somatic mutations (and epimutations) is predominantly governed by the rate of growth and the number of cell divisions per unit time, particularly within the stem cell compartment of the shoot apical meristem (SAM)[33]. Our findings support this view: we observed that experimentally induced acceleration of tree growth correlates with increased rates of epimutations in both the main stem and lateral branches. By focusing on epimutations at CG sites - whose methylation patterns are maintained during DNA replication[44] - these results imply that the increased rates are driven by a higher frequency of cell divisions. Our cell-count assays confirmed this conclusion. However, we acknowledge that these assays were conducted exclusively in hardwood, where SAM-derived cell lineages leave a traceable historical record that can be retrospectively analyzed. As such, these assays provide only an indirect proxy for estimating cell division rates in other tissue types. Directly measuring stem cell division rates in the SAM remains experimentally infeasible due to the technical challenge of live imaging in intact meristems of mature trees[22]. However, wood formation observations (i.e., xylogenesis sampling) could help to at least better grasp the exact cell division dynamics[55].

The "mitotic-rate hypothesis" explains our findings and accounts for broader patterns, such as the negative relationship between generation time and per-year somatic (epi)mutation and substitution rates observed in tree species[23,33,56]. In long-lived species, slower growth and reduced annual cell divisions delay the accumulation of potentially deleterious somatic variants[22,33], affecting both mutation and epimutation rates equally. In contrast, mammalian studies argue that long-lived species have evolved more efficient DNA repair mechanisms[57]. However, this argument is less applicable to CG epimutations, which occur approximately four orders of magnitude more frequently than DNA mutations[10-12,18], show no clear link to DNA damage, and largely

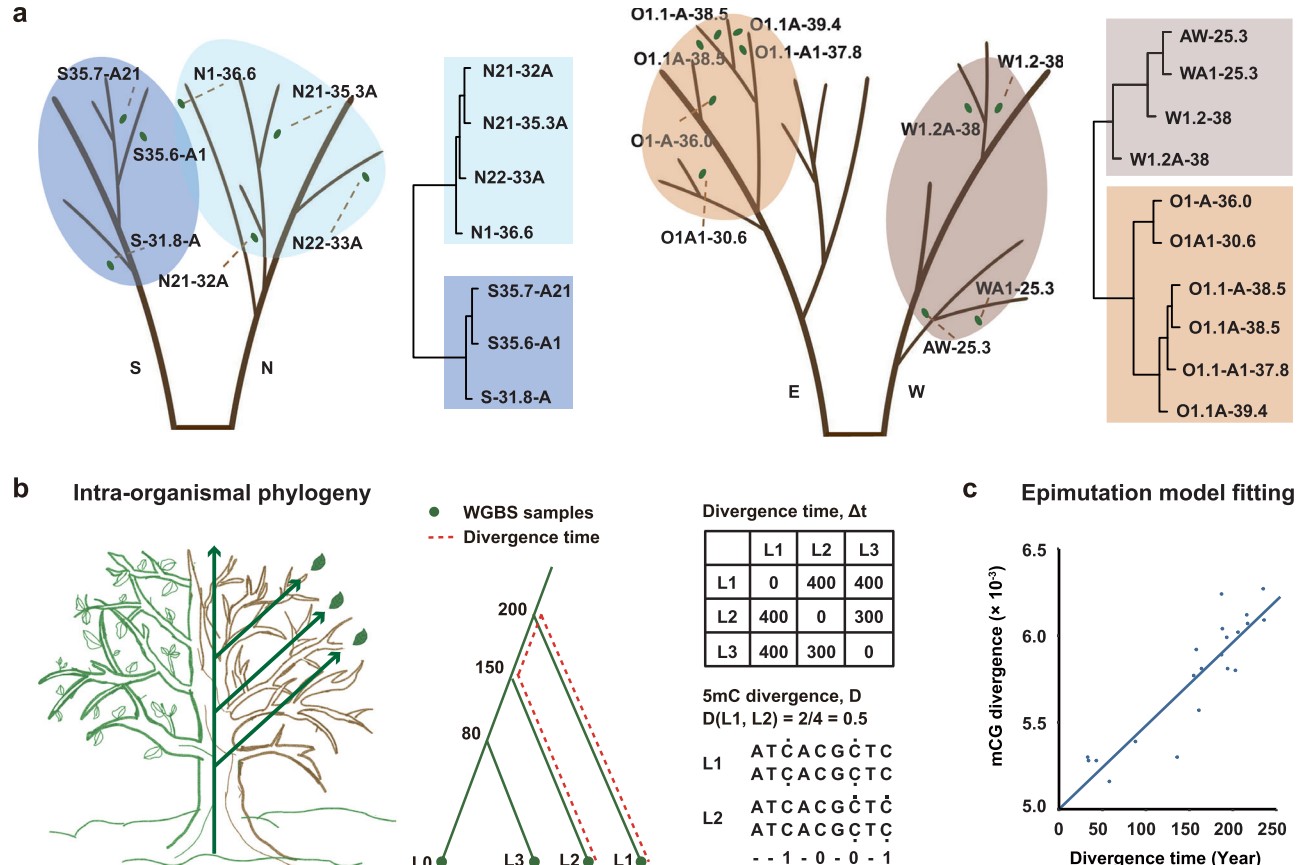

**Fig. 3 | DNA methylation-based clustering of leaf samples recapitulates known tree topology. a** Sample locations of leaves within the branching topologies of trees 109 (left) and 171 (right). Unsupervised sample clustering based on detected SMPs recapitulates the branching topologies of each of the two trees, indicating that SMP analysis identified somatically fixed epimutations. Shaded areas indicate two clusters along the branch directions, enclosing the corresponding samples. The blue shading shows tree 109 samples, with dark and light blue for south- and north-oriented branches, respectively. The brown shading shows tree 171 samples, with dark and light brown for west- and east-oriented branches, respectively. **b** A schematic showing that a tree can be interpreted as an intra-organismal phylogeny. The topology is given and the branch points and branch lengths can be dated by

coring. For simplicity, only three branches are highlighted (L1, to L3) as well as the main stem (L0). Leaf WGBS measurements can be obtained and used to calculate pairwise DNA methylation divergence. Similarly, divergence times (in years) for pairs of leaves can be calculated by tracing back the ages of the branches to their most recent common branch point; here shown for L1 and L2. This can only be done down to the tree's earliest branch point (in this case, Yr = 200) but not to earlier time points. **c** DNA methylation divergence increases as a function of divergence time (Tree 109 as an example. Source data are provided as a Source Data file.). Divergence increases according to a neutral epimutation process, and depends only on the stochastic methylation gain and loss rates at the level of individual CG sites or at the level of regions.

arise from errors by methyltransferases during cell division[19]. Therefore, enhanced DNA repair alone is unlikely to explain the variation in epimutation rates among plants.

As observed in our study, Tree 171 exhibits both a higher rate of cell division and a larger cell size compared to Tree 109. This raises an intriguing possibility that increased cell size and metabolic activity in the faster-growing tree contribute to its elevated epimutation rate. Theoretical models have demonstrated the fact that metabolic rate scales as the three-quarter power of body mass and that ontogenetic growth results from a balance between energy allocated to tissue maintenance and new biomass production[58–60]. This scaling relationship not only underpins organismal growth but also extends to cellular processes, including DNA methylation maintenance. This aligns with recent work showing that somatic (epi)mutation rates scale with generation time in trees according to a power law[23], suggesting an allometric link between life history traits and (epi)genomic maintenance fidelity. Supporting this, metabolic rate itself scales with cell size and growth rate in predictable ways[59], pointing to a potential mechanistic connection between cellular growth, metabolic flux, and methylation fidelity. Bridging these frameworks, our observations are

in line with the hypothesis that life-history strategies in long-lived species may reflect a trade-off between growth and the long-term preservation of (epi)genomic integrity.

An alternative perspective to the "mitotic-rate hypothesis" proposes that the accumulation of somatic variants is determined solely by chronological age rather than developmental time[27]. This "age-related hypothesis"[61] implies that replication-independent mechanisms are the primary drivers of somatic variations. Recent evidence supporting this view comes from a comparison of two tropical tree species, *Shorea laevis* and *Shorea leprosula*, where age, rather than growth rate, appeared to be the key determinant underlying mutation rate differences[27,61]. Although the influence of age on mutation rates has been documented in mammals, for instance, in oocytes - this hypothesis would have made incorrect predictions about the outcome of our study, as fast and slow growing trees had the same age but differing epimutation rates per unit time. Additionally, the hypothesis fails to account for the nearly 35 to 110-fold variation in per-year (epi) mutation rates observed across plant species[23]. If age were the sole factor, per-year rates should be constant across species, with only per-generation rates varying due to differences in lifespans. Instead, the

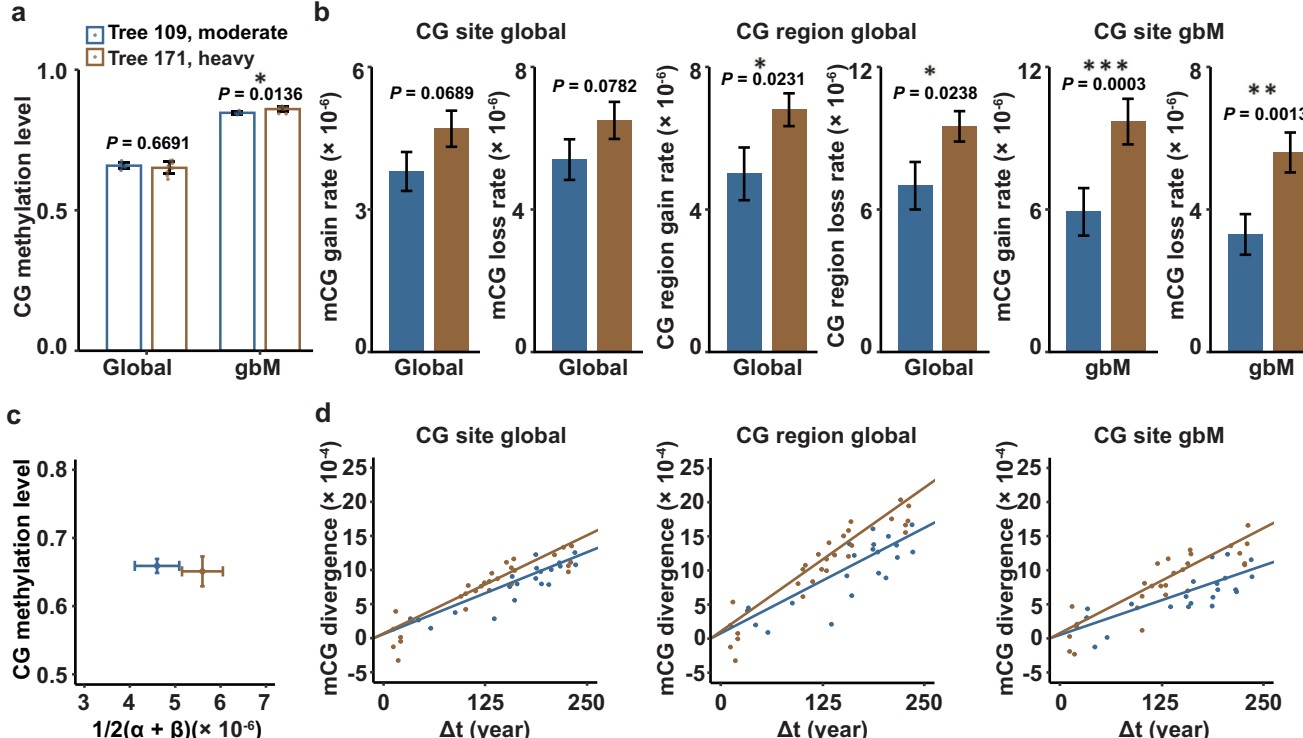

**Fig. 4 | Analysis of somatic epimutation rates. a** Comparison of CG methylation levels reveals no significant differences between Tree 171 (heavily thinned, $N = 10$ biological replicates) and Tree 109 (moderately thinned, $N = 7$ biological replicates). **b** However, Tree 171 has a significantly higher per-site and per-region CG epimutation rate than Tree 109, both genome-wide (globally) as well as within gbM genes. **c** Hence, very similar steady-state CG methylation levels can be accompanied by very different CG epimutation rates. **d** The epimutation rates differences between Tree 171 and Tree 109 are reflected in the fast CG methylation divergence as function of divergence time (in years). Error bars for methylation levels represent the mean ± standard deviation (SD), and error bars for epimutation rates represent the mean ± standard error (SE). Differences in methylation level between two independent trees were determined using a two-sided Wilcoxon rank-sum test. Differences in parameter estimates (gain rate and loss rate) were assessed using a one-sided Welch's *t*-test based on their estimates and standard errors. Full statistical results for all comparisons are provided in Supplementary Data 5. *$P < 0.05$; **$P < 0.01$; ***$P < 0.001$. Source data are provided as a Source Data file.

opposite is observed: when adjusting per-year rates by generation time differences, rates in trees are within a mere ~3-fold of those observed in the short-lived annual plants like *A. thaliana*[23].

This latter observation suggests that the net generational burden of (epi)mutations is evolutionarily constrained across species[23]. This constraint also predicts that accelerated growth is offset by early growth cessation, senescence or mortality, which has some support in data[62–64]. While our study focuses on epimutations, the observation that accelerated growth increases somatic epimutation rates motivates us to hypothesize that accelerated growth may also increase genomic mutation rates. Recent studies have demonstrated that both somatic mutation rates and epimutation rates scale with maturation age in perennial plants such as trees[23,25], suggesting a shared underlying biological mechanism likely associated with the division rates of shoot apical meristem (SAM) cells. In addition, experimental work in fungal systems has provided direct evidence that epigenetic modifications themselves causally influence genomic mutation rates, particularly cytosine methylation in transposable elements, leading to a 15-fold increase in mutation accumulation[65]. Future studies can combine whole-genome sequencing with life-history data to explore how developmental dynamics shape genetic and epigenetic variation in long-lived plants. Given the increasing anthropogenic pressures on forests and climate-driven alterations in tree growth patterns, understanding the trade-offs between growth rate and genome stability has profound implications for forest management and conservation. Our findings contribute insights into the evolutionary constraints shaping plant longevity and genome maintenance.

## Methods

### Study site and sampling design

We relied on the long-term thinning trial Fabrikschleichach 15 (FAB 15) in a European beech (*Fagus sylvatica* L.) forest in Central Europe (Fig. 1a) to obtain trees with exactly the same age, but with strongly divergent growth rates. The region is characterized by a mild climate with an average annual temperature of 7.5 °C and an annual precipitation of 820 mm, with the natural vegetation being submontane European beech-sessile oak forests. FAB 15 is one of the longest continuously measured forest experimental sites worldwide and consists of three differently managed plots, each with a size of ~0.4 ha: unthinned, moderately thinned, and heavily thinned. The beech forest, now over 200 years old, originated from a shelterwood cutting in 1822, and the various treatments and continuous measurements began in 1870/1871. The unthinned plots remained essentially untouched, apart from the occasional removal of dying or dead trees to prevent possible damage to the stand from fungal or insect infestation. In contrast, the managed plots were moderately or heavily thinned to reduce stand density by removing mainly suppressed trees, but also tall trees, especially in the case of the more intense treatment[36]. In total, the managed plots have been thinned 12 times with different intensities since their establishment, resulting in different tree sizes and stem growth rates (Fig. 1b, c). We selected a representative, average sample tree from each thinned plot for further analyses, which was felled and precisely measured. Both selected trees had a particularly low branching point to allow for distinct differences in branch divergence time within the same tree. After felling, branches along the crown

periphery were selected (distributed across lower, middle, and upper crown) and leaves were sampled from the ends of these branches (Fig. 3a). All leaf samples were kept frozen at −80 °C for subsequent DNA methylation analysis. For a correct assignment of branches within the tree architecture, we reconstructed the tree topology and only selected those branches for sampling whose branching path could be clearly traced back. Furthermore, we obtained stem and branch disks from different positions along the stem axis and crown topology to allow for age and stem growth rate estimations. In more detail, a stem disk was extracted from each tree at 1.3 m height, while branch disks from leaf-sampled branches were taken 10 cm before and after each branching position. We followed the sampling methods described in several published studies[66–69] to collect cambium samples at 1.3 m height from moderately thinned and heavily thinned plots. All samples were immediately washed with 70% ethanol and stored at −80 °C for DNA extraction.

### Growth rate analysis

We used the stems disks to measure the annual tree ring widths from four cardinal directions (N, E, S, W; see Fig. 1e) to the nearest of 1/100 mm using a digital positioning table for stem disk measurements (Digitalpositometer, Biritz and Hatzl GmbH, Austria) after sanding the disks with progressively finer sandpaper (up to 800 grit). Tree ring widths were transformed to basal area increments (bai) using the formula $bai_t = \pi * (r_t^2 - r_{t-1}^2)$ (1), where $r$ represents the tree's radius at 1.3 m height and the respective year $t$. We defined the stem growth rate as the average annual basal area increment, considering all four series per stem disk.

### Branch age determination

For determining the age of the different branches, we utilized the derived branch disks, which were sanded to enhance the visibility of annual tree ring borders (see also stem disk preparation). Tree rings of all branch disks (from four cardinal directions per disk) were measured with the digital positioning table Lintab 5 and the software TSAPWin (both Rinntech, Heidelberg, Germany). We visually cross-dated all tree ring series, taking into account distinctive growth patterns across the different samples to ensure the correct dating of the individual tree rings[70]. We counted all tree rings from bark to pith as an estimation of the age of each branch.

### Cell count assays

We relied on wood anatomical methods to derive estimates of annually produced xylem cells for both sample trees. Stem disks were cut with a small circular saw into ~1 cm wide sections, capturing all tree rings from the pith to the bark. For both trees, we used a section without any cracks or damages for further wood anatomical analyses. The cross sections were cut into smaller segments of 3–5 cm before taking transverse micro sections of 10–20 µm thickness with a sliding GSL-1 microtome (Schenkung Dapples, Zürich, Switzerland). Sample preparation followed a protocol by Gärtner & Schweingruber[71]. In more detail, all samples were bleached and washed with distilled water, before they were double-stained with safranin and astrablue for 5 min (mixture of 1:1). Subsequently, the samples were again rinsed with distilled water, ethanol of increasing purity (80%, 96%, and anhydrous ethanol) for dehydration, and xylene before being permanently embedded on a glass slide with Canada balsam and dried out in an oven at 60 °C for 24 h. We used an optical microscope (Zeiss Axio Imager Z2) at × 200 magnification, equipped with an integrated camera (Zeiss Axiocam 305 color) to capture micrographs of wood anatomical features. The micrographs were stitched together using the software ZEN 3.2 blue edition (all three by Carl Zeiss Microscopy Deutschland GmbH, Oberkochen, Germany). Finally, we utilized CAR-ROT, a software based on Deep Convolutional Neural Network (DCNN) algorithms, specifically programmed and trained for quantitative wood anatomy analyses (Supplementary Fig. 4). The software was employed to recognize and segment all types of cells for each ring from the obtained micrographs[72]. We assessed the number of xylem cells per tree ring along a 5000 pixel wide band extending from the start of treatment in 1870 to the felling date in 2020.

### Whole Genome Bisulfite Sequencing analysis

DNA was extracted from cambium tissue using Qiagen's DNeasy Plant Pro Kit (catalog no. 69204, Qiagen) following the manufacturer's protocol with the addition of 100 µl of the solution PS per sample due to high amounts of phenolic compounds in beech species. We used NEBNext Ultra II DNA Library Prep Kit (catalog no. E7103, New England BioLabs) for sequencing library preparation and EZ-96 DNA Methylation-Gold MagPrep (catalog no. D5042, ZYMO) for bisulfite treatment. The protocol involved: i) end repair and 3′ adenylation of sonicated DNA fragments by Covaris R230 (Covaris), ii) NEBNext adaptor ligation, iii) cleanup of libraries with AMPure XP Beads (catalog no. A63881, Beckman Coulter), iv) bisulfite treatment, v) PCR enrichment and index ligation using KAPA HiFi Uracil+ Kit (catalog no. KK2802, Agilent) and NEBNext Multiplex Oligos for Enzymatic Methyl-seq (catalog no. E7140L, New England BioLabs) for bisulfite libraries (12 cycles), vi) final cleanup with AMPure XP Beads. Finally, they were sequenced on a NovaSeq X Plus platform (Illumina) in a paired-end 150 bp format. DNA was extracted from leaves using Qiagen's DNeasy Plant Mini Kit. WGBS libraries were constructed by BGI (Beijing Genomics Institute) and then sequenced on a NovaSeq 6000 platform (Illumina) in a paired-end 150 bp format. After excluding the samples with library construction failure, we had 18 cambium samples and 17 leaf samples. The *Fagus sylvatica* L. reference genome and annotation were used from http://www.beechgenome.net/[73]. The WGBS data were processed with the MethylStar pipeline[74,75]. Specifically, sequencing quality was checked with FastQC v0.11.7, and clean reads were obtained using Trimmomatic v0.39 with parameters: ILLUMINACLIP:-TruSeq3-PE.fa:1:30:9 LEADING:20 TRAILING:20, SLIDINGWINDOW: 4:20 MINLEN:36. We mapped the reads to the reference genome, removed duplicates, and extracted the methylation levels using Bismark v0.19.1 with default parameters. METHimpute was used for cytosine methylation state calling[76]. The mean coverage of samples was about 20X, and the mean mapping rate was about 70% (Supplementary Data 1).

### Identification of gene body methylated genes

We identified gene body methylated (gbM) genes using gbMine (github.com/jlab-code/gbMine)[77]. The software uses the outputs of MethylStar and an annotation (gff3) file. gbMINE provides two flags, --genomicBackground and -- exons, which allow us to identify gbM genes using four different genomic features combinations. We only set -- exons, which means that we only consider exonic cytosines in both the foreground and the background. Using a binomial model, gbMINE classifies a gene as gbM if the proportion of mCG in the exons of that gene is statistically higher than that of exons in the background, while the proportion of mCHG and mCHH is not significantly different. We obtained a set of gbM genes for each sample, and used the union as a liberal set of gbM genes. As a further filter, we calculated the non-CG methylation levels of introns of the union set, which generated a distribution of intronic non-CG methylation levels for the candidate gbM gene. Finally, we only retained gbM genes whose intronic non-CG methylation levels were less than the median of that distribution.

### SMP and DMR calling

DMR calling was performed using jDMR[12,78] (github.com/jlab-code/jDMRgrid). Briefly, we divided the genome into windows of 100 bp each. Within these windows, we calculated the number of methylated and total cytosines, which were used as input for Methimpute[76], a finite state Hidden Markov Model (HMM) with binomial emission densities.

We employed Methimpute's three-state HMM, where each 100 bp window was classified as methylated, unmethylated, or intermediate. The intermediate state calls are designed to capture somatic epiheterozygotes, which are only visible in WGBS data in the form of "intermediate" methylation levels. This approach yielded a matrix representing the methylation state of each region and sample (Supplementary Data 2). jDMR queries this matrix to identify methylation state switches between samples within each 100 bp window (e.g., unmethylated to intermediate methylated) and defines these as DMRs. A similar strategy was employed to call SMPs, with the window size being reduced to single CG sites. Although our SMP and DMR analysis was based on bulk leaf methylome data, jDMR's HMM approach is robust to measurement noise arising from the cell layer-specificity in which somatic variations often originate in the SAM[79,80]. This becomes evident from the clustering results in Fig. 3a.

### Genome-wide methylation divergence analysis

Cambium samples were obtained from four moderately and from three heavily thinned plots. Three samples were taken from each tree: two neighboring replicate samples (R1 and R2) from the same side of the stem and one sample (S) from the polar opposite side. We calculated the methylation difference between two replicates (R1 vs. R2, RR), as well as the methylation difference between each replicate and the sample from the opposite side (R1 vs. S, R2 vs. S). Firstly, we calculated methylation divergence for each cytosine site $i$ (posteriorMax ≥ 0.99), for example, R1 vs. S, $D_i = |M(R1, i) - M(S, i)|$ (2) and then, we averaged the methylation difference at each cytosine site to obtain the genome-wide methylation difference,

$$D = \frac{1}{n} \sum_{i=1}^{n} D_i \quad (3)$$

Differentially methylated positions (DMPs) are employed as proxies for the methylation divergence for better interpretability. Cohen's $d$ is defined as $d_C = \frac{D_1 - D_2}{s}$ (4), with the pooled standard deviation,

$$s = \sqrt{\frac{(n_1 - 1) s_1^2 + (n_2 - 1) s_2^2}{n_1 + n_2 - 2}} \quad (5)$$

To more accurately quantify methylation divergence induced by different thinning intensities, we normalized the opposite-side divergence by subtracting the baseline variation observed between same-side replicates. In other words, the methylation difference between two sides of the stem is defined as the "opposite sides methylation difference" minus the "same-side replicates methylation difference". We then calculated the fold change between treatments based on these adjusted values. Using the mean values shown in Fig. 2b: 0.005221033 (replicates), 0.005485005 (moderately), and 0.005919238 (heavily), the fold change was calculated as: (0.005919238 − 0.005221033) / (0.005485005 − 0.005221033) = 2.64. Based on this, we stated that "methylation divergence between samples from opposite sides of the stems was 2.64-fold higher in trees from the heavily thinned plot compared to trees from the moderately thinned plot."

### Tree topology construction

To construct the tree's topological structure, we first divided the fallen tree into segments and systematically marked each branch with identifiers. These markings enabled precise tracking and mapping of the branches, allowing us to accurately reconstruct the tree's topology. This method ensured that we could reliably determine the spatial relationships between branches, ensuring an accurate representation of the tree's structure, as shown in Fig. 3a.

### Epimutation rate estimation

We estimated somatic epimutation rates using AlphaBeta[17], a computational method that infers the spontaneous epimutation rate from WGBS data. AlphaBeta requires methylation data (the output of MethylStar) and pedigree data as input (Supplementary Data 4). Following Shahryary et al.[17] and Hofmeister et al.[13], we treated the tree branching structure as a pedigree of somatic cell lineages, where the leaves represent the lineage end-points. AlphaBeta calculates the methylation divergence ($D$) and divergence time ($\Delta t$) between all pairs of leaf samples. The model parameters are estimated using numerical nonlinear least squares optimization. We estimated epimutation rates under a neutral epimutation model (AlphaBeta's ABneutralSOMA model).

### Reporting summary

Further information on research design is available in the Nature Portfolio Reporting Summary linked to this article.

## Data availability

All WGBS raw data used for this article have been deposited in the NCBI SRA under accession PRJNA1215776. The data underlying Figs. 1–4 and Supplementary Figs. 1, 2 are provided on Zenodo under https://doi.org/10.5281/zenodo.17350136. Source data are provided with this paper.

## Code availability

All software and code used for the analyses in this study were developed by our lab, and are available from GitHub under https://github.com/jlab-code/MethylStar; https://github.com/jlab-code/gbMine; https://github.com/jlab-code/jDMRgrid; https://github.com/jlab-code/AlphaBeta.

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

## Acknowledgements

This work was partly funded by the Bundesministerium für Bildung und Forschung (Project: epiSOMA). We would like to thank the Bavarian State Institute of Forestry (Forest Protection Department) and the Ecophysiology group at TUM for their support with technical equipment and the Bavarian State Forests (BaySF) for their support in setting up and maintaining the underlying long-term experiments. We thank W. Wanney, L. Schlegel, and R. Bhardwaj for helping collect cambium samples. G. Schmied acknowledges further funding from the Bavarian State Ministry of Food, Agriculture and Forestry (Project: beechGPT). M. Zhou holds a fellowship from the China Scholarship Council (CSC NO. 202204910009).

## Author contributions

M.Z. and G.S. performed all data analysis; M.B., E.U., T.H., A.T.C., G.S., H.P., and F.J. performed field work; B.T., M.Z., and G.R. performed the experiment; R.H., I.K., M.C., M.S., R.L.P., R.J.S., and A.C. provided advice; F.J. and H.P. supervised and directed this project; M.Z., G.S. and F.J. wrote the manuscript with input from all authors.

## Funding

## Competing interests

The authors declare no competing interests.
