## [Transparent Peer Review file · Nature Communications]

Accelerated growth increases the somatic epimutation rate in trees

Corresponding Author: Professor Frank Johannes

Version 0:

Reviewer comments:

Reviewer #1

(Remarks to the Author)

In this manuscript the authors investigate the relationship between growth rate and somatic epimutation rates in trees, using European beech (*Fagus sylvatica*) as a model. The researchers leveraged a 150-year-old thinning experiment to compare trees subjected to different thinning intensities, which resulted in varying growth rates. They found that thinning-induced growth acceleration was accompanied by increased cell division rates in the main stems, as confirmed by cell count assays. Trees with higher growth rates exhibited significantly higher somatic epimutation rates in both main stems and lateral branches. Although the results are interesting, the analysis of epimutation rates is based on a small number of trees (two representative trees for detailed analysis), which may limit the generalizability of the findings.

Here are several concerns and suggestions:

1. Line 99-111, Please provide a figure to show the difference in stem growth rates, diameter breast height, and crown size between the treatments at a population level.
2. How representative are the two selected trees of their respective treatments?
Do they have a close genetic relationship or genotypes?
3. Line 136-136, are they significantly different?
4. Fig2. What's the full name of DMP? How do you calculate the CG/CHG divergence? How about CHH divergence?
5. Fig2B, the variation from R1 vs. R2 is too big, and can't tell which dots are from which treatment. Color coding them? Since the variation between neighboring replicate samples is big, not sure one sample from the polar opposite side is enough.
6. Fig. How do you construct the topology tree?
7. What distribution of these gained or lost epimutations (or DMR) across the genome? Are they randomly mutated? What kind of genes were under epimutations? Do these mutations affect gene expression levels? Are these epimutations neutral or deleterious?
8. Could differences in stress responses (due to thinning) independently affect methylation? Other factors, like local environment, soil, or the genotypic background of the tree itself may also affect methylation.
9. Mean coverage was ~20X—was this sufficient for low-frequency epimutation detection?
10. Does the accelerated growth affect the genomic mutation rate?
11. Missing title in Table S2

Reviewer #2

(Remarks to the Author)

The study by Zhou et al. explored the impact of tree growth rates on epimutation frequencies. To this end, the authors took advantage of an experimental plot where beech trees have been grown for decades under distinct thinning strategies. Trees in heavily thinned regions grow faster compared to those in less thinned areas. The authors exploited this difference to study epimutations (changes in DNA methylation) associated with accelerated growth.

First, the authors demonstrated that faster growth is associated with increased cell division (and also with increased cell expansion). They then measured global methylation divergence between faster growing and “control” trees grown in less thinned areas by comparing vascular tissues from opposite sides of the trunk. The analysis revealed that faster-growing trees exhibit higher levels of methylation divergence, specifically in the CG context, but not in the CHG context.

Subsequently, the authors performed an intra-organismal experiment in which methylation divergence was calculated as a function of divergence over time. For this, they compared methylomes of distinct leaves whose positions in the tree could be tracked and also determined the ages of the branches on which these leaves were located. The results indicated that leaves from the faster-growing tree showed increased methylation divergence, supporting their previous finding when comparing vascular tissues.

This is a very interesting manuscript using a unique genetic material that allowed to report a clear positive correlation between plant growth rate and epimutation rate. While there is no direct evidence proving that increased cell division is responsible for the elevated epimutation rate, these findings align well with the previously proposed “mitotic-rate hypothesis”.

Comments/Suggestions:

- An interesting point is that cells in the faster-growing tree are also larger. Could this phenotype contribute to the increase in epimutation rate? Perhaps increased metabolic during extended cell growth activity also alters the deposition/removal of DNA methylation. This question is difficult to address with the current experimental model but perhaps could be added to the discussion if the authors think it could be possible.
- Fig. 2B, C: Please provide a more detailed description in the Methods section of how global mC divergence was calculated. Also, indicate in the figure legend where the “n” values come from, and why certain dots are shown in red.
- Fig. 2B, C: Please indicate the statistical significance of the differences between measurements in the plots.
- Fig. 2C: Were the CG and CHG divergence values calculated across all C positions in the genome? Please describe this more clearly. Additionally, what is the rationale for showing CHG but not CHH methylation for non-CG contexts? Would including CHH methylation offer further insights or reinforce the CHG findings?
- Related to Fig. 2: The authors state that “methylation divergence between samples from opposite sides of the stems was 2.64-fold higher in trees.” Please clarify how this number was calculated.
- The authors present global CG divergence between trees and leaves. Are these loss and gains enriched in specific genomic regions or associated with genes with particular features (e.g., gene length, intron number,...), or are the epimutations randomly distributed across the coding genome?
- Fig. 3C: Please specify which datasets were used to generate this plot.
- The authors mention: “Removing the low-quality sequencing samples, we had 18 cambium samples and 17 leaf samples.” Please clarify which specific samples were excluded from the analyses.
- Supplementary Table 1: Please include information indicating which sample IDs correspond to the CB and CC samples.
- On page 5, please correct the typo: “METHYLTRANSFERASE” should be “METHYLTRANSFERASE.”

Version 1:

Reviewer comments:

Reviewer #1

(Remarks to the Author)

I appreciate that the authors addressed most of my questions, but I still have some concerns, see below

1. Fig. 2b.c Please provide P-values and statistical method. What do the red dots indicate?

2. For my question 7

Can you make a figure to show the distribution of these DMRs?

The authors addressed my questions by citing some papers. Is there any study work on the same species as you did here? If not, it's unclear whether you can draw any conclusions.

3. For my question 9

In the citations, 20X coverage was used in Arabidopsis, not sure whether it's suitable for European beech.

I didn't find the paper by Shahryary et al. 2020 in the Reference

Reviewer #2

(Remarks to the Author)

I would like to thank the authors for their comments and the corrections they provided. I am satisfied with their responses and the revised version. I have two comments regarding the revised manuscript that could be considered:

- Line 421: Please consider revising the sentence: "To more accurately quantify different thinning intensities induced methylation divergence, ..."

- Thank you for reporting the genomic distribution of CG divergence in both trees in Fig. 1 of the rebuttal. I wonder why this data was not included in the manuscript, considering both reviewers asked the same question. It is up to the authors to decide whether to include it, but I think it is an informative piece of data. If yes, please add to the legend that these are CG-context DMRs.

REVIEWER COMMENTS

Reviewer #1 (Remarks to the Author):

In this manuscript the authors investigate the relationship between growth rate and somatic epimutation rates in trees, using European beech (*Fagus sylvatica*) as a model. The researchers leveraged a 150-year-old thinning experiment to compare trees subjected to different thinning intensities, which resulted in varying growth rates. They found that thinning-induced growth acceleration was accompanied by increased cell division rates in the main stems, as confirmed by cell count assays. Trees with higher growth rates exhibited significantly higher somatic epimutation rates in both main stems and lateral branches. Although the results are interesting, the analysis of epimutation rates is based on a small number of trees (two representative trees for detailed analysis), which may limit the generalizability of the findings.

Here are several concerns and suggestions:

1. Line 99-111, Please provide a figure to show the difference in stem growth rates, diameter breast height, and crown size between the treatments at a population level.

We thank the reviewer for this suggestion. This analysis is now shown in the new Supplementary Fig. 1.

2. How representative are the two selected trees of their respective treatments? Do they have a close genetic relationship or genotypes?

The two selected trees, Tree 109 and Tree 171, are representative individuals of their respective treatments. As shown in Fig. 1c and in Supplementary Fig. 1, their growth trajectories, tree dimensions and other characteristics fall within the intermediate range of trees in their respective plots, indicating they reflect typical patterns for their thinning treatment. Furthermore, all trees within the FAB 15 experiment originated from natural regeneration following shelterwood cuttings in 1822. This silvicultural practice promotes regeneration from a common seed source under similar ecological conditions. While the exact genotypes of Tree 109 and Tree 171 have not yet been determined, there is no indication of artificial selection or planting, and all trees in the plots are presumed to stem from the same local gene pool. Therefore, although we cannot confirm a close genetic relationship between the two selected trees, it is reasonable to assume they are representative members of a naturally regenerated, locally adapted beech population subjected to the respective long-term thinning treatments.

3. Line 136-136, are they significantly different?

We observed a significantly greater CG methylation divergence in the heavily thinned plot compared to the moderately thinned plot, as assessed by a t-test (p -value < 0.05). We have included the statistical significance of the differences in the plots of Fig. 2b and c. Although not all comparisons reached statistical significance, the relatively large Cohen's *D* effect sizes support an increasing trend from RR to moderately thinned treatment to heavily thinned treatment, which supports our central hypothesis. Please refer to updated Fig. 2 for details.

4. Fig2. What's the full name of DMP? How do you calculate the CG/CHG divergence? How about CHH divergence?

DMP stands for Differentially Methylated Position. We are now defining this term in the Method section. To calculate CG and CHG divergence, we took the following steps: For each tree disk, we calculated the methylation difference between two replicates located on the same side (R1 vs. R2), as well as the methylation difference between each replicate and the sample from the opposite side (R1 vs. S, R2 vs. S). To obtain a

measure of genome-wide methylation difference between samples, we followed our previously published approach (see e.g. van der Graaf et al. 2015, Hazarika et al. 2022, Shahryary et al. 2020). For each cytosine site i , we calculated the methylation difference between the two samples (e.g., R1 and S): $D_i = |M(R1, i) - M(S, i)|$, and then took the average over all covered cytosines in the genome:

$$D = \frac{1}{n} \sum_{i=1}^n D_i$$

We thank the reviewer for bringing up this question. We are now detailing this approach in the Methods section.

5. Fig2B, the variation from R1 vs. R2 is too big, and can't tell which dots are from which treatment. Color coding them? Since the variation between neighboring replicate samples is big, not sure one sample from the polar opposite side is enough.

We appreciate the reviewer's suggestion. Indeed, the variation between neighboring replicate samples (e.g., R1 vs. R2) appears relatively large. However, when looking at the mean divergence differences, there is a clear trend showing that R1 vs. R2 (RR) are, on average, much less divergent. Although the small sample sizes may limit statistical significance, the relatively large Cohen's D effect sizes support an increasing trend from RR to moderately thinned treatment to heavily thinned treatment. Please refer to updated Fig. 2 for details.

6. Fig. How do you construct the topology tree?

To construct the tree's topological structure, we first divided the fallen tree into segments and systematically marked each branch with identifiers. These markings enabled precise tracking and mapping of the branches, allowing us to accurately reconstruct the tree's topology. This method ensured that we could reliably determine the spatial relationships between branches, ensuring an accurate representation of the tree's structure.

7. What distribution of these gained or lost epimutations (or DMR) across the genome? Are they randomly mutated? What kind of genes were under epimutations? Do these mutations affect gene expression levels? Are these epimutations neutral or deleterious? Overall, Tree 171 exhibits a higher number of CG-context DMRs compared to Tree 109, yet both share a similar distribution pattern across genomic regions (Fig. 1). The majority of CG-DMRs are located in non-coding regions (intergenic) and transposable elements (TEs), which is a typical pattern in plants (Schmitz et al. 2013; Niederhuth et al. 2016; Zhang et al. 2018). A substantial number of DMRs are found in CDS regions. Intronic regions also contain many DMRs. In contrast, DMRs in exonic regions are relatively sparse. Regarding the potential impact of the identified epimutations on gene expression levels, although we did not have gene expression data (such as RNA-seq) in this study, prior work has shown that gene body methylation (gbM) changes, which constitute the majority of somatic epimutations, typically have little to no effect on gene expression levels (Zhang et al. 2018; Bewick et al. 2019; Hofmeister et al. 2020). This provides strong support that the epimutations detected here are largely functionally neutral, and it ensures that the accumulation of epimutations is associated with cell divisions rather than driven by natural (i.e. cell lineage) selection.

8. Could differences in stress responses (due to thinning) independently affect methylation? Other factors, like local environment, soil, or the genotypic background of the tree itself may also affect methylation.

We thank the reviewer for raising this important point. Our experimental design aimed to minimize the influence of these variables. All three 0.4 ha plots lie in the immediate vicinity to each other on loamic Luvisols, displaying very similar soil characteristics. All

sampled trees were selected randomly within the different treatments, avoiding a systematic bias. While localized microclimatic variation cannot be entirely ruled out, particularly given that higher thinning intensity reduces standing stock and may slightly alter environmental conditions, such differences are considered to be a direct consequence of the treatment itself rather than an independent confounding factor. Finally, all trees originated from natural regeneration following a shelterwood cut, so they derive from the same locally adapted gene pool; there was no artificial planting or selection of distinct genotypes in any plot. Although we have not yet performed molecular genotyping on Trees 109 and 171, their shared origin and randomized selection make it reasonable to assume no systematic genotypic differences among treatments. Thus, based on our randomized selection and the uniform soil conditions, we minimize other factors that might affect methylation. Moreover, the “thinning” itself only involved clearing dead wood or dying trees from the plots, and is not expected to impose stress on the surrounding trees. Nonetheless, we appreciate the reviewer's thoughts here: The correlative link between increased rates of growth, cell division and epimutations should be further investigated in future studies to pin point the precise causal relationship. However, the careful observation that such a correlative link exists is an important first step that our study has taken.

9. Mean coverage was ~20X—was this sufficient for low-frequency epimutation detection?

From our experience with similar experimental setups (Becker et al. 2011; Graaf et al. 2015; Schmitz et al. 2022), a 20× coverage is sufficient for the detection of most epimutations. Additionally, our method for estimating rates is relatively robust to ascertainment error as we explicitly model the error in the estimation procedure (see Shahryary et al. 2020).

10. Does the accelerated growth affect the genomic mutation rate?

We did not assess genomic mutation rates in this study, as our focus was specifically on epimutations. Evaluating whether accelerated growth also affects the genomic mutation rate would require whole-genome resequencing data, which was beyond the scope of our current work. However, having shown that epimutation rates are closely linked to cell division rates, we hypothesize that accelerated growth may influence genomic mutation rates similarly. Recent studies have demonstrated that both somatic mutation rates and epimutation rates scale with maturation age in perennial plants such as trees (Hanlon et al. 2019; Johannes 2025), suggesting a shared underlying biological mechanism likely associated with the division rates of shoot apical meristem (SAM) cells.

We thank the reviewer for bringing up this point. We have now included a brief discussion of this hypothesis in the discussion section.

11. Missing title in Table S2

Thanks for pointing out the missing title in Table S2. We have now added a title to the table, and the revised version is included in the updated files.

Reviewer #2 (Remarks to the Author):

The study by Zhou et al. explored the impact of tree growth rates on epimutation frequencies. To this end, the authors took advantage of an experimental plot where beech trees have been grown for decades under distinct thinning strategies. Trees in heavily thinned regions grow faster compared to those in less thinned areas. The authors exploited this difference to study epimutations (changes in DNA methylation) associated with accelerated growth.

First, the authors demonstrated that faster growth is associated with increased cell division (and also with increased cell expansion). They then measured global methylation divergence between faster growing and “control” trees grown in less thinned areas by comparing vascular tissues from opposite sides of the trunk. The analysis revealed that faster-growing trees exhibit higher levels of methylation divergence, specifically in the CG context, but not in the CHG context.

Subsequently, the authors performed an intra-organismal experiment in which methylation divergence was calculated as a function of divergence over time. For this, they compared methylomes of distinct leaves whose positions in the tree could be tracked and also determined the ages of the branches on which these leaves were located. The results indicated that leaves from the faster-growing tree showed increased methylation divergence, supporting their previous finding when comparing vascular tissues.

This is a very interesting manuscript using a unique genetic material that allowed to report a clear positive correlation between plant growth rate and epimutation rate. While there is no direct evidence proving that increased cell division is responsible for the elevated epimutation rate, these findings align well with the previously proposed “mitotic-rate hypothesis”.

We thank this reviewer for the positive assessment of our work.

Comments/Suggestions:

- An interesting point is that cells in the faster-growing tree are also larger. Could this phenotype contribute to the increase in epimutation rate? Perhaps increased metabolic during extended cell growth activity also alters the deposition/removal of DNA methylation. This question is difficult to address with the current experimental model but perhaps could be added to the discussion if the authors think it could be possible. The potential link between cell size, metabolic activity, and epimutation rate is indeed an interesting hypothesis. This discussion aligns well with the recent observation that both somatic mutation and epimutation rates show a powerlaw scaling with maturation age in trees (Johannes 2025), suggesting that there is an allometric link between life history traits and (epi)genomic maintenance fidelity. It is therefore of theoretical interest to explore how the scaling connects with the allometric laws that were previously found to link cellular metabolic rate, body size, and growth (West et al. 2002). Our current study cannot dissect these links, but our observations are consistent with this hypothesis and highlight a potential direction for future work.

We thank the reviewer for the insightful suggestion. We have now included a brief discussion of this hypothesis in the discussion section.

- Fig. 2B, C: Please provide a more detailed description in the Methods section of how global mC divergence was calculated. Also, indicate in the figure legend where the “n” values come from, and why certain dots are shown in red.

Thanks for the suggestion. We have revised the manuscript accordingly. A more detailed description has been added to the Methods section. Additionally, we have updated the figure legend to indicate the source of the “n” values and to explain why certain dots are shown in red.

- Fig. 2B, C: Please indicate the statistical significance of the differences between measurements in the plots.

Thanks, we have included the statistical significance of the differences in the plots of

Fig. 2b, c.

- Fig. 2C: Were the CG and CHG divergence values calculated across all C positions in the genome? Please describe this more clearly. Additionally, what is the rationale for showing CHG but not CHH methylation for non-CG contexts? Would including CHH methylation offer further insights or reinforce the CHG findings?

We thank the reviewer for this helpful comment. The CG and CHG methylation divergence values in Fig. 2c were calculated genome-wide, based on all covered (posteriorMax ≥ 0.99) cytosines in the respective sequence contexts. We have clarified this in the Methods section of the revised manuscript. Consistent with previous work both in somatic systems (Hofmeister et al. 2020; Ibañez et al. 2023) and transgenerational studies (Quadrana and Colot 2016; Zheng et al. 2017; Li et al. 2020), we found no time-dependent accumulation in CHH epimutations. This is likely due to the fact that the methylation in CHH context is mainly targeted by de novo methylation pathways that prevent the accumulation of stable gains and losses over generations (Johannes and Schmitz, 2019).

- Related to Fig. 2: The authors state that “methylation divergence between samples from opposite sides of the stems was 2.64-fold higher in trees.” Please clarify how this number was calculated.

Thanks for the question. We have added more detailed explanations in the Methods section. As shown in Fig. 2b, the first box represents methylation divergence between replicate samples on the same side of the stem (R1 vs. R2). The second and third boxes represent divergence between samples from opposite sides of the stem under moderately and heavily thinned treatments, respectively.

To more accurately quantify different thinning intensities induced methylation divergence, we normalized the opposite-side divergence by subtracting the baseline variation observed between same-side replicates. In other words, the methylation difference between two sides of the stem is defined as the “opposite sides methylation difference” minus the “same-side replicates methylation difference”. We then calculated the fold change between treatments based on these adjusted values. Using the mean values shown in Fig. 2b: 0.005221033 (replicates), 0.005485005 (moderately), and 0.005919238 (heavily), the fold change was calculated as: $(0.005919238 - 0.005221033) / (0.005485005 - 0.005221033) = 2.64$.

Based on this, we stated that “methylation divergence between samples from opposite sides of the stems was 2.64-fold higher in trees from the heavily thinned plot compared to trees from the moderately thinned plot.”

- The authors present global CG divergence between trees and leaves. Are these loss and gains enriched in specific genomic regions or associated with genes with particular features (e.g., gene length, intron number,...), or are the epimutations randomly distributed across the coding genome?

Overall, Tree 171 exhibits a higher number of CG-context DMRs compared to Tree 109. Yet, both share a similar distribution pattern across genomic regions (Fig. 1). The majority of CG-DMRs are located in non-coding regions (intergenic) and transposable elements (TEs), which is a typical pattern in plants (Schmitz et al. 2013; Niederhuth et al. 2016; Zhang et al. 2018). A substantial number of DMRs are found in CDS regions. Intronic regions also contain many DMRs. In contrast, DMRs in exonic regions are relatively sparse.

- Fig. 3C: Please specify which datasets were used to generate this plot.

Thanks for the comment. This information has now been clarified in the revised figure legend.

- The authors mention: “Removing the low-quality sequencing samples, we had 18

cambium samples and 17 leaf samples.” Please clarify which specific samples were excluded from the analyses.

This was a misstatement; it should refer to samples that failed library construction. We have corrected this.

- Supplementary Table 1: Please include information indicating which sample IDs correspond to the CB and CC samples.

We thank the reviewer for pointing this out. We have now added information to Supplementary Data 1.

- On page 5, please correct the typo: “METHYLTRANSFERASE” should be “METHYLTRANSFERASE.”

Thanks for the correction. We have corrected “METHYLTRANSFERASE” to “METHYLTRANSFERASE” in the revised manuscript.

Reference

1. Schmitz, R., Schultz, M., Urich, M. et al. Patterns of population epigenomic diversity. *Nature* 495, 193–198 (2013).
2. Niederhuth, C. E., Bewick, A. J., Ji, L. et al. Widespread natural variation of DNA methylation within angiosperms. *Genome Biol.* 17, 194 (2016).
3. Zhang, H., Lang, Z. & Zhu, J. K. Dynamics and function of DNA methylation in plants. *Nat. Rev. Mol. Cell Biol.* 19, 489–506 (2018).
4. Bewick, A. J., Zhang, Y., Wendte, J. M., Zhang, X. & Schmitz, R. J. Evolutionary and experimental loss of gene body methylation and its consequence to gene expression. *G3 Genes|Genomes|Genetics* 9, 2441–2445 (2019).
5. Hofmeister, B. T., Denkena, J., Colomé-Tatché, M. et al. A genome assembly and the somatic genetic and epigenetic mutation rate in a wild long-lived perennial *Populus trichocarpa*. *Genome Biol.* 21, 259 (2020).
6. Becker, C., Hagmann, J., Müller, J. et al. Spontaneous epigenetic variation in the *Arabidopsis thaliana* methylome. *Nature* 480, 245–249 (2011).
7. van der Graaf, A., Wardenaar, R., Neumann, D. A. et al. Rate, spectrum, and evolutionary dynamics of spontaneous epimutations. *Proc. Natl Acad. Sci. USA* 112, 6676–6681 (2015).
8. Schmitz, R. J., Marand, A. P., Zhang, X. et al. Quality control and evaluation of plant epigenomics data. *Plant Cell* 34, 503–513 (2022).
9. Hanlon, V. C. T., Otto, S. P. & Aitken, S. N. Somatic mutations substantially increase the per-generation mutation rate in the conifer *Picea sitchensis*. *Evol. Lett.* 3, 348–358 (2019).
10. Johannes, F. Allometric scaling of somatic mutation and epimutation rates in trees. *Evolution* 79, 1–5 (2025).
11. West, G. B., Woodruff, W. H. & Brown, J. H. Allometric scaling of metabolic rate from molecules and mitochondria to cells and mammals. *Proc. Natl Acad. Sci. USA* 99 (Suppl 1), 2473–2478 (2002).
12. Noé Ibañez, V., van Antrop, M., Peña-Ponton, C., Milanovic-Ivanovic, S., Wagemaker, C. A. M., Gawehns, F. & Verhoeven, K. J. F. Environmental and genealogical effects on DNA methylation in a widespread apomictic dandelion lineage. *J. Evol. Biol.* 36, 663–674 (2023).
13. Quadrana, L. & Colot, V. Plant transgenerational epigenetics. *Annu. Rev. Genet.* 50, 467–491 (2016).
14. Zheng, X., Chen, L., Xia, H. et al. Transgenerational epimutations induced by multi-generation drought imposition mediate rice plant's adaptation to drought condition. *Sci. Rep.* 7, 39843 (2017).

15. Li, J., Yang, D. L., Huang, H. et al. Epigenetic memory marks determine epiallele stability at loci targeted by de novo DNA methylation. *Nat. Plants* 6, 661–674 (2020).
16. Johannes, F. & Schmitz, R. J. Spontaneous epimutations in plants. *New Phytol.* 221, 1253–1259 (2019).

Fig. 1 The distribution of differentially methylated regions (DMRs) for Tree 109 and Tree 171.

REVIEWER COMMENTS – updated

Reviewer #1 (Remarks to the Author):

I appreciate that the authors addressed most of my questions, but I still have some concerns, see below

We would like to thank the reviewer for insightful suggestions, which have greatly improved the quality and completeness of our manuscript.

1. Fig. 2b.c Please provide P-values and statistical method. What do the red dots indicate? We thank the reviewer for this very helpful suggestion. We have updated Figure 2b, c to include the exact P-value for the significant result, and provided the statistical method in the legend, and full statistical results, including non-significant differences, are provided in Supplementary Data 5. The red dots deviate from the overall data distribution and may be potential outliers; however, they do not affect the overall results or conclusions, so they are retained in the figure.

2. For my question 7

Can you make a figure to show the distribution of these DMRs?

The authors addressed my questions by citing some papers. Is there any study work on the same species as you did here? If not, it's unclear whether you can draw any conclusions.

We thank the reviewer for the follow-up question. To our knowledge, there is currently no published study on CG-DMR distribution in European beech. Nevertheless, our analysis provides a description of somatic CG-DMRs in this species, offering valuable insight into the genomic distribution of epimutations. While direct comparisons with previous work in the same species are not possible, the observed patterns are consistent with general trends reported in other plants (as cited in our initial response), supporting the robustness and interpretability of our data.

3. For my question 9

In the citations, 20X coverage was used in *Arabidopsis*, not sure whether it's suitable for European beech.

Thanks for your comment. The cited study utilized 20x sequencing depth in *Arabidopsis*, and 20x coverage has also been successfully applied in other plant species, for instance, in *Zea mays* and *Pinus tabulaeformis* (Xu et al. 2020; Li et al. 2023). Increasing sequencing depth will improve sensitivity, but a 20x coverage is sufficient for the detection of most epimutations.

I didn't find the paper by Shahryary et al. 2020 in the Reference

Shahryary, Y., Symeonidi, A., Hazarika, R. R. et al. AlphaBeta: computational inference of epimutation rates and spectra from high-throughput DNA methylation data in plants. *Genome Biol.* **21**, 260 (2020).

Reference

1. Xu, G., Lyu, J., Li, Q. et al. Evolutionary and functional genomics of DNA methylation in maize domestication and improvement. *Nat Commun* 11, 5539 (2020).
2. Li, J., Han, F., Yuan, T. et al. The methylation landscape of giga-genome and the epigenetic timer of age in Chinese pine. *Nat Commun* 14, 1947 (2023).
3. Shahryary, Y., Symeonidi, A., Hazarika, R. R. et al. AlphaBeta: computational inference of epimutation rates and spectra from high-throughput DNA methylation data in plants. *Genome Biol.* **21**, 260 (2020).

Reviewer #2 (Remarks to the Author):

I would like to thank the authors for their comments and the corrections they provided. I am satisfied with their responses and the revised version. I have two comments regarding the revised manuscript that could be considered:

We thank the reviewer for valuable suggestions, which have greatly improved the clarity and quality of our manuscript.

- Line 421: Please consider revising the sentence: "To more accurately quantify different thinning intensities induced methylation divergence, ..."

Thanks for your suggestion. We have revised the sentence for clarity and readability. The updated version now reads:

To more accurately quantify methylation divergence induced by different thinning intensities.

- Thank you for reporting the genomic distribution of CG divergence in both trees in Fig. 1 of the rebuttal. I wonder why this data was not included in the manuscript, considering both reviewers asked the same question. It is up to the authors to decide whether to include it, but I think it is an informative piece of data. If yes, please add to the legend that these are CG-context DMRs.

Thanks for your suggestion. We agree that the CG-context DMRs are informative data, and have incorporated the distribution of CG-DMRs into Supplementary Figure 3. We think this will give readers a more intuitive understanding of our data.